



# Global seismic energy scaling relationships based on the type of faulting.

**Quetzalcoatl Rodríguez-Pérez[1], F. Ramón Zúñiga[2]**

[1] Dirección de Desarrollo Científico, Dirección de Cátedras, Consejo Nacional de Ciencia y Tecnología, Mexico City, Mexico.
[2] Centro de Geociencias, Universidad Nacional Autónoma de México, Juriquilla, Querétaro, Mexico.

Correspondence: Quetzalcoatl Rodríguez-Pérez (*quetza@geociencias.unam.mx*)

**Abstract.** We derived scaling relationships for different seismic energy metrics for earthquakes with $M_W > 6.0$ from 1990 to 2022. The seismic energy estimations were derived with two methodologies, the first based on the velocity flux integration and the second based on finite-fault models. In the first case, we analyzed 3331 reported seismic energies derived by integrating far-field waveforms. In the latter methodology, we used the total moment rate functions and the approximation of the overdamped dynamics to quantify seismic energy from 231 finite-fault models ($E_{mrt}$, and $E_O$, $E_U$, respectively). Both methodologies provide compatible energy estimates. The radiated seismic energies estimated from the slip models and integration of velocity records are also compared for different focal mechanisms by deriving converting scaling relations among the different energy types. Additionally, the behavior of radiated seismic energy ($E_R$), energy-to-moment ratio ($E_R/M_0$), and apparent stress ($\tau_\alpha$) for different rupture types at a global scale is examined by considering depth variations of mechanical properties, such as seismic velocities and rock densities, and rigidities. For this purpose, we used a 1-D global velocity model. In agreement with previous studies, our results exhibit a robust variation of $\tau_\alpha$ with the focal mechanism. These parameters are, on average largest for strike-slip earthquakes, followed by normal-faulting events, with the lowest values for reverse earthquakes for hypocentral depths < 180 km. On the contrary, at depths in the range of 180 - 240 km, $\tau_\alpha$ for reverse earthquakes is higher than for normal-faulting events. Regarding the behavior of apparent stress with depth, our results agree with



the existence of a bimodal distribution with two depth intervals where the apparent stress is maximum
for normal-faulting earthquakes. Finite-fault energy estimations also support focal mechanism
dependence of apparent stress, but only for shallow earthquakes (Z < 30 km). The population of slip
distributions used was too small to conclude that finite-fault energy estimations support the dependence
of average apparent stress on rupture type at different depth intervals.
**1 Introduction**
The radiated seismic energy ($E_R$) is a crucial source parameter that accounts for the size of an
earthquake. The seismic energy is also a valuable parameter for understanding the dynamics of the
rupture, especially in the case of large and complex earthquake sources (Venkataraman and Kanamori,
2004a; Convers and Newman, 2011). The radiated seismic energy is considered the main contribution
to the total wave energy radiated by an earthquake (Boatwright and Choy, 1986). The most common
approach to calculating $E_R$ requires the integration of radiated energy flux in velocity-squared
seismograms (Haskell, 1964; Thatcher and Hanks, 1973; Boatwright, 1980; Kanamori et al., 1993;
Boatwright and Choy, 1986; Singh and Ordaz, 1994; Choy and Boatwright, 1995; Pérez-Campos and
Beroza, 2001). In order to recover the $E_R$ of an event, the seismic records have to be corrected for
propagation path and source effects such as attenuation, site effects, geometric spreading, radiation
pattern, and directivity. In calculating seismic energy, information on the Earth's structure is required
since $E_R$ needs to be measured over a broad range of distances. Inaccurate information on the Earth's
structure results in uncertainties in energy estimations, particularly at higher frequencies
(Venkataraman and Kanamori, 2004a). Furthermore, previous studies showed that estimates of $E_R$
based on regional and teleseismic data might differ by as much as a factor of 10 for the same
earthquake (Singh and Ordaz, 1994).



Choy and Boatwright (1995) reported a focal mechanism dependence on $E_R$. Later this observation was
confirmed by Pérez-Campos and Beroza (2001), showing that the mechanism dependence is not as
strong as reported previously. The degree of dependence of seismic energy on the focal mechanism is
affected by several factors that bias the estimate (e.g., uncertainties in the corner frequency, geometrical
spreading, hypocentral depth, and focal mechanism) (Pérez-Campos and Beroza, 2001). This
dependence can be expressed in terms of the apparent stress ($\tau_\alpha = \mu\, E_R/M_0$, where $\mu$ is the rigidity, Wyss
and Brune, 1968), energy to moment ratio ($E_R/M_0$), or slowness parameter ($\Theta = \log_{10}(E_R/M_0)$, Newman
and Okal, 1998). Previous studies showed that strike-slip events have the highest apparent stress ($\tau_\alpha =$
0.70 Mpa), followed by normal-faulting and thrust earthquakes with 0.25 and 0.15 MPa, respectively
(Pérez-Campos and Beroza, 2001). On the other hand, some authors have observed that $E_R/M_0$ ratio is
different for different types of earthquakes, particularly in subduction zones. For example, tsunami
earthquakes have the smallest $E_R/M_0$ ratio ($7 \times 10^{-7} - 3 \times 10^{-6}$), interplate and downdip events have a
slightly larger ratio ($5 \times 10^{-6} - 2 \times 10^{-5}$), and intraplate and deep earthquakes have $E_R/M_0$ ratios similar
to crustal earthquakes ($2 \times 10^{-5} - 3 \times 10^{-4}$) (Venkataraman and Kanamori, 2004a). The origin of the
focal mechanism dependence is unclear, but it has been proposed that the stress drop is the cause of this
dependence of the radiated seismic energy on the type of faulting (Pérez-Campos and Beroza, 2001).

Other approaches have also been used to calculate seismic energy, such as those based on finite-fault
models (Ide, 2002; Venkataraman and Kanamori, 2004b; Senatorski, 2014). Ide (2002) calculated the
radiated energy using an expression based on slip and stress on the fault plane. Energy estimates from
this method tend to be smaller by about a factor of 3 compared with the integrating far-field waveforms
method. Venkataraman and Kanamori (2004b) used a formula for the energy radiated seismically from
a finite source as a function of the time-dependent seismic moment $M_0(t)$ and the properties of the
medium. Here, the moment rate function derived from kinematic inversion is used to calculate the $E_R$.





On the other hand, Senatorski (2014) used an overdamped dynamics approximation for estimating the
radiated seismic energy. The accuracy of this method depends on the rupture history. This approach
provides two energy parameters: 1) The finite-fault overdamped dynamics approximation ($E_O$) and, 2)
the energy obtained from the averaged finite-fault model ($E_U$). In both cases, the seismic energy
depends on the slip, rupture time, and seismic moment. According to Senatorski (2014), in most cases,
the radiated seismic energy estimated by integrating digital seismic waveforms ($E_R$) is in the following
range: $E_U < E_R < E_O$. Several seismic energy observations have been calculated and compiled in
different catalogs in the last two decades. In this study, we reexamine the possible dependence of
seismic energy on the focal mechanism with an additional classification based on the type of rupture,
considering pure and oblique mechanisms separately. We also investigate the potential influence of
focal mechanism on the derived estimates of radiated seismic energy from finite-fault models.
Additionally, we explored the relationship between depth and the variables $E_R/M_0$ and $\tau_\alpha$. Furthermore,
we established conversion relationships between various types of energy estimates. These findings play
a crucial role in enhancing our understanding of the rupture processes associated with different types of
earthquakes.

**2 Data and methods**
**2.1 Data**
We retrieved and classified focal mechanism solutions from the global CMT catalog (Ekström et al.,
2012) using a ternary diagram based on the Kaverina et al. (1996) projection. This approximation
classifies focal mechanism into seven classes of earthquakes: 1) normal (N); 2) normal – strike-slip (N-
SS); 3) strike-slip – normal (SS-N); 4) strike-slip (SS); 5) strike-slip – reverse (SS-R); 6) reverse –
strike-slip (R-SS); and 7) reverse (R) (Fig. 1). For implementing fault-plane classification, we used the
software FMC developed by Álvarez-Gómez (2019). Additionally, we used radiated seismic energy





data and finite-fault models reported by the Incorporated Research Institutions for Seismology (IRIS)
and the United States Geological Survey (USGS), respectively. To have homogeneity in the analyzed
data, we do not include seismic energy observations and finite-fault models from other sources to avoid
bias. IRIS reported automated $E_R$ solutions for global earthquakes with an initial magnitude above $M_W$
6.0. We studied 3331 events worldwide during the period April 1990 – October 2022 (Fig. 2). Results
include broadband energy solution (frequency band in the interval of 0.5 – 70 s) from vertical-
component seismograms recorded at teleseismic distances ($25° \leq \Delta \leq 80°$) (Convers and Newman,
2011; Hutko et al., 2017). Finite-fault models are determined with a kinematic inversion based on the
wavelet domain (Ji et al., 2002). The procedure jointly inverts body and surface waves on a fault plane
aligned with focal mechanism estimates from USGS W-phase or gCMT solutions. We used 231 finite-
fault models from 1990 to 2022 (Fig. 2). After classifying the events, we determined scaling
relationships for the reported seismic energies and analyzed the behavior of the $E_R/M_0$ ratio and $\tau_\alpha$. The
seismic energy was also determined using finite-fault models with the techniques described in the
following section to know if there is a difference in estimates related to the faulting type. Seismic
velocities and rock densities were taken from the ak135-F velocity model (Kennett et al., 1995;
Montagner and Kennett, 1995); rigidity was calculated as $\mu = \rho\beta^2$.

**2.2 Methods**
**2.2.1 Radiated seismic energy derived from seismic waves**
In the following, we described the procedure to calculate $E_R$ implemented by IRIS. Radiated energies
used in this study were calculated with the method of Boatwright and Choy (1986) as implemented by
Convers and Newman (2011). Using velocity seismograms of the *P*-wave group (consisting of
*P+pP+sP* phases), the energy is calculated at teleseismic distances. The seismic energy flux from the *P*-
wave group ($\varepsilon_{gP}$) is calculated from the velocity spectrum ( $\dot{u}(\omega)$ ) as:




$$\varepsilon_{gP} = \frac{\rho(z)\alpha(z)}{\pi} \int\limits_0^\infty |\dot{u}(\omega)|^2 \exp\left(\omega t_\alpha^*\right) d\omega \quad , \tag{1}$$


where $\rho(z)$ and $\alpha(z)$ are the density and $P$-wave velocity at the source depth $(z)$, and the exponential
term $t_\alpha^*$ corrects for anelastic attenuation. Subsequently, the energy flux is corrected for geometrical
spreading, radiation pattern, and partitioning between $P$ and $S$ waves. The radiated seismic energy at a
given station is calculated as:

$$E_R^P = 4\pi \langle F^P \rangle^2 \left(\frac{R^P}{F^{gP}}\right)^2 \varepsilon_{gP} \quad , \tag{2}$$


where $\langle F^P \rangle^2$ is the mean radiation pattern coefficient for $P$-waves, $R^P$ is the geometrical spreading
factor of $P$-waves, $F^{gP}$ is the generalized radiation pattern coefficient for the $P$-wave group.

$$\left(F^{gP}\right)^2 = \left(F^P\right)^2 + \left(PP \, F^{pP}\right)^2 + \frac{2\alpha(z)}{3\beta(z)} q \left(CSP \, F^{sP}\right)^2 \quad , \tag{3}$$


where $\beta(z)$ is the $S$-wave velocity at the source depth, $C$ is the correction for wavefront sphericity, $F_p$,
$F_{pP}$, and $F_{sP}$ are radiation pattern coefficients for the $P$, $pP$, and $sP$ waves, respectively (Aki and
Richards, 1980). The parameter $q$ represents the relative partitioning between $S$ and $P$ waves (using $q =$
15.6, Boatwright and Fletcher, 1984). $PP$ and $SP$ are the reflection coefficients for the $pP$ and $sP$ wave
phases at the free surface. Finally, the radiated seismic energy obtained from the $P$-wave or $S$-wave
groups can be estimated with the formulae $E_R = (1 + q)E_R^P = (1 + 1/q)E_R^S$. For each event, the final



assigned seismic energy is the average for all the stations used.

**2.2.2 Radiated energy estimations from finite-fault slip models**
Senatorski (2014) introduced a method to estimate energy parameters derived from kinematic slip
models. In this method, the radiated seismic energy is expressed in terms of slip velocities using an
overdamped dynamics approximation (Senatorski, 1994; 1995). The method provides two energy
parameters: 1) the overdamped dynamics energy approximation ($E_O$) and 2) the uniform model energy
estimation ($E_U$). The accuracy of the overdamped dynamics solutions depends on the rupture history.
Senatorski (2014) showed that in most cases, $E_U < E_R < E_O$. The energy parameter $E_O$ is calculated as:

$$E_O = \frac{1}{2\,\beta(z)} \sum_i M_0^i V^i \quad ,$$    (4)

where $\beta(z)$ is the shear wave velocity at the source depth and $M_0^i$ is the seismic moment released at
the $i$-th fault segment. $V^i$ is given by $V^i = D^i / t_R^i$, and $D^i$, and $t_R^i$ are the slips and risetimes at the $i$-th
segment, respectively. The averaged finite-fault model estimation assumes uniform slip ($\bar{D}$), and
slip velocity ($V = \bar{D}/T$), so

$$E_U = \frac{1}{2\,\beta(z)} M_0 V \quad ,$$    (5)

where $M_0$ is the total seismic moment, and $T$ is the rupture duration.




### 2.2.3 Radiated energy estimates based on moment rate functions of slip models

The radiated seismic energy can also be calculated through moment rate functions of finite-fault models (Haskell, 1964; Aki and Richards, 1980; Rudnicki and Freud, 1981; Venkataraman and Kanamori, 2004b). By ignoring the contribution from *P*-waves, which accounts for less than 5 % of the total radiated energy, the radiated energy derived from moment rate functions ($E_{mrt}$) can be written as (Venkataraman and Kanamori, 2004b):

$$E_{mrt} = \frac{1}{10\,\pi\rho(z)\,\beta^5(z)} \int_0^\infty \left| \ddot{M}(t)_0 \right|^2 dt \quad ,$$

where $\rho(z)$ and $\beta(z)$ are the density and *S*-wave velocity, respectively, at the source depth, and $\ddot{M}(t)_0$ is the derivative of the moment rate function ( $\dot{M}_0(t)$ ) estimated from a finite-fault model.

### 3 Results

We used different methods to quantify the radiated seismic energy. Table 1 shows the calculated scaling relationships for $E_R$ for each energy method and type of faulting. Figs. 3, 4, 5, and 6 display the energy scaling relations derived from the velocity flux integration ($E_R$), overdamped dynamics energy approximation ($E_O$), the uniform model energy estimation ($E_U$), and moment rate function methods ($E_{mrt}$), respectively. Our results showed some disparities in the calculated radiated seismic energies obtained with different techniques or data types. When comparing $E_R$ with the other methods to estimate seismic energy, we find that the lowest average difference factors are for $E_O$ estimates, ranging from 0.28 to 0.77 (Fig. 7). Conversely, mean difference factors can be as high as 20 for $E_U$ estimations (Fig. 8). Average difference factors exhibit intermediate values for $E_{mrt}$ calculations, fluctuating from 1.53 to 3.27 (Fig. 9). Regarding the rupture type, reverse earthquakes have the highest dispersion, but



they have the most significant number of observations (Figs. 7 to 9). Conversion relationships between
$E_R$ and $E_O$, $E_U$, and $E_{mrt}$ are presented in Table 2, which may be helpful when considering either method
of estimation.

In terms of the $E_R/M_0$ ratio, our results showed that SS, SS-N, and SS-R events have the highest mean
values ($3.06 \times 10^{-5} < E_R/M_0 < 3.75 \times 10^{-5}$) (Fig. 10). R-SS earthquakes have a slightly lower mean ratio
($E_R/M_0 = 2.87 \times 10^{-5}$) (Fig. 10). Average $E_R/M_0$ ratio fluctuates from $2.31 \times 10^{-5}$ to $2.37 \times 10^{-5}$ for N-SS
and N events, respectively (Fig. 10). On the other hand, the lowest values of $E_R/M_0$ are related to R
earthquakes ($E_R/M_0 = 1.70 \times 10^{-5}$) (Fig. 10). Most of the rupture types present a differentiated behavior
of $E_R/M_0$ in terms of depth with the existence of two clusters, above and below about 300 km depth
(Fig. 11). In contrast, strike-slip earthquakes demonstrate a distinct pattern, with the majority of $E_R/M_0$
observations concentrated at depths shallower than 50 km (Fig. 11). Furthermore, at shallow depths, the
radiated energy-to-moment ratio shows large variability compared to observations of deep earthquakes
(Fig. 11).

Previous studies have provided evidence that mean $\tau_\alpha$ estimates can be obtained using regression
models, specifically through the equation $\log_{10} E_R = \log_{10} M_0 + b$ with $\tau_\alpha = \mu 10^b$, supporting the focal
mechanism dependence of $E_R$ (Choy and Boatwright, 1995; Pérez-Campos and Beroza, 2001). To test
that this dependence persists with depth, we conducted regressions every 30 km of depth considering
variations of μ and at least ten observations. First, we evaluated reported seismic energy observations
based on the velocity flux integration method (Table 3). Our results for average apparent stress agree
with previous studies where $\tau_\alpha$ follows the following behavior (R-SS, R) < (N-SS, N) < (SS, SS-N, SS-
R) in the range of 0 − 180 km (Table 3). On the contrary, $\tau_\alpha$ is higher for R events than for N
earthquakes at depths from 180 to 240 km (Table 3). At depths higher than 240 km, only N events were





obtained under the assumptions considered. In Table 3, we summarized results for all the depth
intervals showing the mean values and their 95% log-normal geometric spread.

Our results also showed that N and N-SS events exhibit a bimodal distribution of $\tau_\alpha$ with depth (Fig.
12). The most significant values of $\tau_\alpha$ occur in two depth ranges of approximately 40 – 60 km and 580 –
650 km, where maximum apparent stresses approach 8 and 16 MPa, respectively (Fig. 12). N-SS, R, R-
SS, SS-N, and SS-R events also showed two maximum values of $\tau_\alpha$ ranging from 7 to 11 MPa and 9 to
15 MPa for shallow and deep earthquakes, respectively (Fig. 12). For SS events, there is only one depth
range over which $\tau_\alpha$ for strike-slip earthquakes shows maxima. In this case, the highest values of $\tau_\alpha$ are
found in the deeper depth range from 50 to 100 km ($\tau_\alpha \sim$ 12 MPa) (Fig. 12). On the other hand, the
average apparent stress estimates based on the finite-fault models exhibit a similar dependence on the
focal mechanism than those obtained with the velocity flux integration method at shallow depths (Z <
30 km) (Table 4). Regressions showed that $\tau_\alpha$ follows the following behavior R < N < (SS, SS-R) for $E_U$
and $E_{mrt}$ estimations (Table 4). In contrast, $E_O$ showed no clear dependence of $\tau_\alpha$ with the focal
mechanism (Table 4). Due to the constraint of at least ten observations (slip distributions) for each 30
km depth interval, we could not analyze the dependence of $\tau_\alpha$ on the type of faulting at a deeper depth.

**4 Discussion**
In this study, we analyzed radiated seismic energy and parameters that measure the amount of energy
per unit of the moment, such as the apparent stress and the energy-to-moment ratio (also known as
scaled energy or apparent strain), considering their respective particularities. The advantage of using $\tau_\alpha$
is that it can be related to other stress processes associated with the seismic rupture, such as the stress
drop. On the other hand, many finite-fault models of the spatiotemporal slip history for moderate and
large earthquakes exist. From these models, important information can be extracted, such as fault



dimensions (Mai and Beroza, 2000), static stress drop (Ripperger and Mai, 2004), or radiated seismic
energy (Ide, 2002; Senatorski, 2014). When using finite-fault models to determine $E_R$, it is necessary to
consider that they usually explain low-frequency seismic waves. However, the higher-frequency wave
contribution is necessary for calculating the total radiated seismic energy. This issue brings differences
among finite-fault energy estimates and those from integrating far-field waveforms.

Furthermore, finite-fault seismic energy estimations are strongly affected by event location, the number
of available data, faulting parameterization, and velocity structure. The degree of discrepancy between
the finite-fault energy estimates ($E_{mrt}$, $E_O$, and $E_U$) with respect to the velocity flux integration method
($E_R$) is variable among the different types of seismic energy. For example, the moment rate functions
are relatively robustly determined by teleseismic data, while rupture dimensions are strongly affected
by model parameters (Ye et al., 2016). This may explain why the average difference factor ($E_R/E_U$) is
greater than the $E_R/E_{mrt}$ factor (Figs. 8 and 9). Another source of discrepancies in finite-fault energy
calculations comes from the spatial and temporal smoothing in resolving the kinematic slip distribution
and the rupture velocity assigned. Errors associated with the assumptions are tough to quantify as they
propagate into the energy estimates in complex ways.

Our results agree with previous estimates of $E_O$ and $E_U$, confirming that $E_R \in (E_U, E_O)$ for most
earthquakes. The overdamping approximation ($E_O$) can be used to characterize the heterogeneity of the
rupture process. Senatorski (2014) states that if the ratio $E_O/E_R$ is < 0.4, the rupture can be represented
as a simple dislocation rupture. $E_O/E_R > 1$ is expected in the case of heterogeneous rupture processes.
On the other hand, some of the suggested explanations for the observation that $E_O > E_R$ are: 1) the
finite-fault slip models require refinement; 2) the seismic energy estimations require correction for
directivity, modified attenuation factors, or sites effects; and 3) some other factors are not considered in



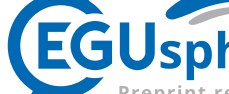

the calculations such as the fact that the energy dissipation is not taken into account by the planar faults
(Senatorski, 2014).

The radiated seismic energy scaled by seismic moment is an essential characterization of earthquake
dynamics. The low $E_R/M_0$ of reverse events is associated with tsunami events being compatible with the
results of previous studies (Newman and Okal, 1998; Venkataraman and Kanamori, 2004a; Convers
and Newman, 2011; Ye et al., 2016). Our results showed that $E_R/M_0$ has a large scatter from 6 x $10^{-7}$ to 2
x $10^{-4}$ for all the rupture types, but no evident magnitude dependence (Fig. 10). One of the reasons for
the dispersion of $E_R/M_0$ is that it depends on many seismogenic properties of the source region (Fig.
10). As a consequence, $E_R/M_0$ varies significantly in different tectonic environments and deep
conditions such as pressure and temperature (Fig. 11). Even within the same tectonic environment,
$E_R/M_0$ has significant variations, as has been reported by Plata-Martínez et al. (2019) in the Middle
American Trench, where variations in $E_R/M_0$ are associated with heterogeneities along the trench, such
as asperities patches. The different types of earthquakes have differences in the frequency content of
the seismic energy released.

Venkataraman and Kanamori (2004a) reported that $E_R/M_0$ is in the range of 5 x $10^{-6}$ – 2 x $10^{-5}$ for
interplate and downdip earthquakes, which are mainly consistent with reverse and normal faulting. Our
results showed that the average values of $E_R/M_0$ for R and N events are 1.70 x $10^{-5}$ and 2.37 x $10^{-5}$,
respectively, and both values are within the interval defined by Venkataraman and Kanamori (2004a).
The $E_R/M_0$ ratio for deep earthquakes varies from 2.0 x $10^{-5}$ to 3.0 x $10^{-4}$ (Venkataraman and Kanamori,
2004a). We found that $E_R/M_0$ for deep earthquakes of all types of rupture is in the interval of 2 x $10^{-6}$ –
2 x $10^{-4}$ but with a predominance of 1.0 x $10^{-5}$ > $E_R/M_0$ (Fig. 11). Despite the $E_R/M_0$ scatter, our results
depict a general trend for the average values of $E_R/M_0$, which can be expressed as R < (N, N-SS, R-SS)





$<$ (SS, SS-R, SS-N) (Fig. 10), a similar tendency was reported by Convers and Newman (2011) where
$E_R/M_0$ follows R $<$ N $<$ SS.

Our results support the previously reported focal mechanism dependence of $E_R$ (Choy and Boatwright,
1995; Pérez-Campos and Beroza, 2001; Convers and Newman, 2011) but narrow the range.
Examination of mean $\tau_\alpha$ with various focal mechanisms and at different depths has been done for
different earthquake sizes and tectonic settings. We identified the largest values of apparent stress for
strike-slip events, intermediate values for normal-faulting events, and lowest for reverse-faulting events
in the depth interval of 0 – 180 km (Table 3). On the other hand, our results showed that at depths
between 180 and 240 km, $\tau_\alpha$ for reverse earthquakes is higher than for normal-faulting events. This can
be explained, for example, in subduction zones, deep reverse earthquakes occur in the lower part of the
slab, where they are subjected to significantly large compressive stresses. A precise characterization of
the depth dependence of $\tau_\alpha$ remains unclear at depths greater than 240 km. In Table 3, we present and
compare our results for $\tau_\alpha$, supporting the observation of the dependence of $E_R$ on the type of faulting.
The origin of this focal dependence is unclear, but it has been raised that it reflects a mechanism-
dependent difference in stress drop (Pérez-Campos and Beroza, 2001). It can be highlighted with an
alternative definition for the apparent stress assuming that the dynamic and static stress drops are
roughly equivalent. Then $\tau_\alpha$ can be expressed as $\tau_\alpha = (\eta_R \, \Delta\sigma)/2$, where $\eta_R$ is the seismic efficiency, and
$\Delta\sigma$ is the stress drop (Convers and Newman, 2011). Allmann and Shearer (2009) provided additional
information to support the role of stress drop on the dependency of apparent stress with the type of
faulting. They found a dependence of median stress drop on the focal mechanism with a factor of 3–5
times higher stress drops for strike-slip events and two times higher stress drops for intraplate events
compared to interplate events.



Nevertheless, other interpretations of the apparent stress variation are related to the mechanical
properties of the rock, such as the reduction of rigidity in shallow subduction environments or
increment in lithostatic pressure if no change in regional rigidity is assumed (Convers and Newman,
2011). In fact, the variation of such estimates concerning expected spatial variations in rigidity is an
issue that still needs attention. Choy and Kirby (2004) also suggested that $\tau_\alpha$ can be related to fault
maturity. For example, lower stress drops are needed to reach rupture in mature faults. On the contrary,
earthquakes generated at immature faults (low cumulative displacement) radiate more energy per unit
of seismic moment. Regarding the behavior of $\tau_\alpha$ with depth, our results agree with the existence of a
bimodal distribution with two depth intervals where the apparent stress is maximum for normal-
faulting earthquakes, as reported by Choy and Kirby (2004). We also found that almost all types of
faulting (SS-N, SS-R, R-SS, R, N-SS, and N) show two depth ranges where the stress is maximum, but
in the case of normal-faulting earthquakes, it is very well defined. On the other hand, almost all strike-
slip earthquakes show a single interval of depths where the apparent stress is maximum (Fig. 12).
Earthquakes with an oblique focal mechanism show a mixed behavior of $\tau_\alpha$, as is the case of the SS-N
and SS-R events that present similar characteristics to normal and reverse earthquakes in terms of the
depth distribution of $\tau_\alpha$.

In terms of the spatial distribution of $E_R$ and $\tau_\alpha$ (Figs. S1 to S14), the highest values of $\tau_\alpha$ for N events
are located at the border between the Nazca and South American plates, the Eurasian and Philippine
plates, the Indo-Australian and Pacific plates, the Philippine and Pacific plates, and the Pacific and
North American plates (in the Alaska region) (Fig. S1). Regarding the seismic energy of earthquakes,
the regions where the most energetic earthquakes have occurred concur with the aforementioned areas,
with the addition of the border between the Cocos and North American plates (Fig. S2). The high $\tau_\alpha$
normal-faulting events are associated with regions of intense deformation, such as a sharp slab bending



or zones where opposing slabs collide (Choy and Kirby, 2004). At shallow depths (Z < 35 km), high-$\tau_\alpha$
events are related to the beginning of the subduction beneath the overriding plate (Choy and Kirby,
2004). Our results support the observation that the average apparent stress of intraslab normal-faulting
events is considerably higher than the average $\tau_\alpha$ of interplate thrust-faulting earthquakes reported by
Choy and Kirby (2004) (Figs. S1 and S5).

In the case of R earthquakes, the highest values of $E_R$ and $\tau_\alpha$ are in the limit of the Eurasian and
Philippine plates, the Nazca and South American plates, the Philippine and Pacific plates, the Indo-
Australian and Pacific plates, and, the Eurasian and Indo-Australian plates (Figs. S5 and S6). In
contrast, strike-slip events with the highest values of $E_R$ and $\tau_\alpha$ are on the border between the African
and Eurasian plates (in Türkiye), the Eurasian and Indo-Australian plates, the Philippine and  Eurasian
plates, the Indo-Australian and Pacific plates (in New Zealand), and the Caribbean and South American
plates (Figs. S13 and S14). We have found that several SS earthquakes are located in the oceanic
lithosphere at depths < 50 km. Many of the SS events with high $\tau_\alpha$ are located near the plate-boundary
triple junctions where there are high rates of intraplate deformation, as previously reported by Choy
and McGarr (2002).

Finally, when using seismic energy estimates based on finite-fault models ($E_O$ and $E_{mrt}$), a clear
dependence of the average apparent stress with the focal mechanism is observed at shallow depths (Z <
30 km) (Table 4). For example, using $E_U$ and $E_{mrt}$, the average $\tau_\alpha$ follows R < N < (SS-R, SS). If $E_O$ is
used, the mean apparent stress exhibits similar values for SS-R, N, and R events (Table 4). However,
the lack of a significant number of observations for some types of earthquakes makes it challenging to
evaluate the use of finite-fault models to determine apparent stress. Despite these limitations, the
methods used to estimate the seismic energy based on finite-fault models are a quick alternative to





calculate a range of energy variation once a slip distribution is obtained.

**5 Conclusion**
We studied the radiated seismic energy, energy-to-moment ratio, and apparent stress for a different type
of faulting. Our data relies on different methodologies employing the velocity flux integration and
finite-fault models to determine the seismic energy. The approach based on slip distributions involved
the utilization of two techniques: 1) total moment rate functions and 2) overdamped dynamics
approximation. We analyzed 3331 energy observations derived from integrating far-field waveforms.
On the other hand, we used 231 finite-fault models. The energy estimates are consistent with each
other, with the maximum average difference factor for $E_U$ estimates followed by $E_{mrt}$ and $E_O$,
respectively. The estimated energy differences are within the margin reported in the literature, which
can reach a factor higher than 10. The methods used to estimate seismic energy based on finite fault
models are an easily implemented alternative that gives results compatible with the seismic record
integration technique, given the larger uncertainties of these methods. We also derived scaling
relationships for the different types of energies and conversion relations.

In terms of the behavior of the $E_R/M_0$ ratio, our results showed a high scatter without a clear
dependence on magnitude. Like previous studies, we observe a robust variation of $E_R/M_0$ with the type
of faulting, which can be expressed as R < (N, N-SS, R-SS) < (SS, SS-R, SS-N). Our $E_R/M_0$ estimates
for deep earthquakes are also consistent with reported values. By analyzing the average apparent stress,
our results also support the previously reported focal mechanism dependence of $E_R$ at depths ranging
from 0 to 180 km. We found that normal-faulting events have intermediate values of $\tau_\alpha$ between strike-
slip and reverse events using the energy flux integration approach in agreement with previous studies.
On the other hand, $\tau_\alpha$ for reverse earthquakes is higher than for normal-faulting events at depths

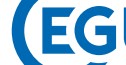

between 180 and 240 km. In contrast, a clear focal mechanism dependence is observed when finite-
fault methods are used to estimate the mean apparent stress at shallow depths (Z < 30 km). This study's
population of slip distributions was too small to conclude that finite-fault energy estimations support
the mechanism dependence of average apparent stress at different depths. There are two depth ranges
over which apparent stress for SS-N, SS-R, R-SS, R, N-SS, and N earthquakes shows maxima.
Earthquakes with an oblique focal mechanism show a mixed behavior of energy parameters since it has
common characteristics of two types of faults; in some cases, one of them predominates over the other.

Code availability. Generic Mapping Tools (GMT5) is available at http://gmt.soest.hawaii.edu/, last
access: 19 June 2023.  FMC is available at https://github.com/Jose-Alvarez/FMC, last access: 19 June

393 2023.


Data availability. Radiated seismic energy data are acquired from the IRIS Data Services Products:
EQEnergy (https://ds.iris.edu/ds/products/eqenergy/). Focal mechanisms are taken from Global CMT
catalog (https://www.globalcmt.org/). Finite-fault models are acquired from the USGS earthquake
catalog (https://earthquake.usgs.gov/earthquakes/search/).

Author contributions. QRP designed the idea, developed the methodology and performed the
preliminary analyses. QRP and FRZ discussed and analyzed the results and wrote the paper.

Competing interests. The authors declare that they have no conflict of interest.

Acknowledgments. Quetzalcoatl Rodríguez-Pérez was supported by the Mexican National Council for
Science and Technology (CONACYT) (Cátedras program - project 1126).

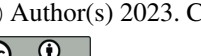



Financial support. This research has been supported by the CONACYT (grant no. Catedras program,
project 1126).

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





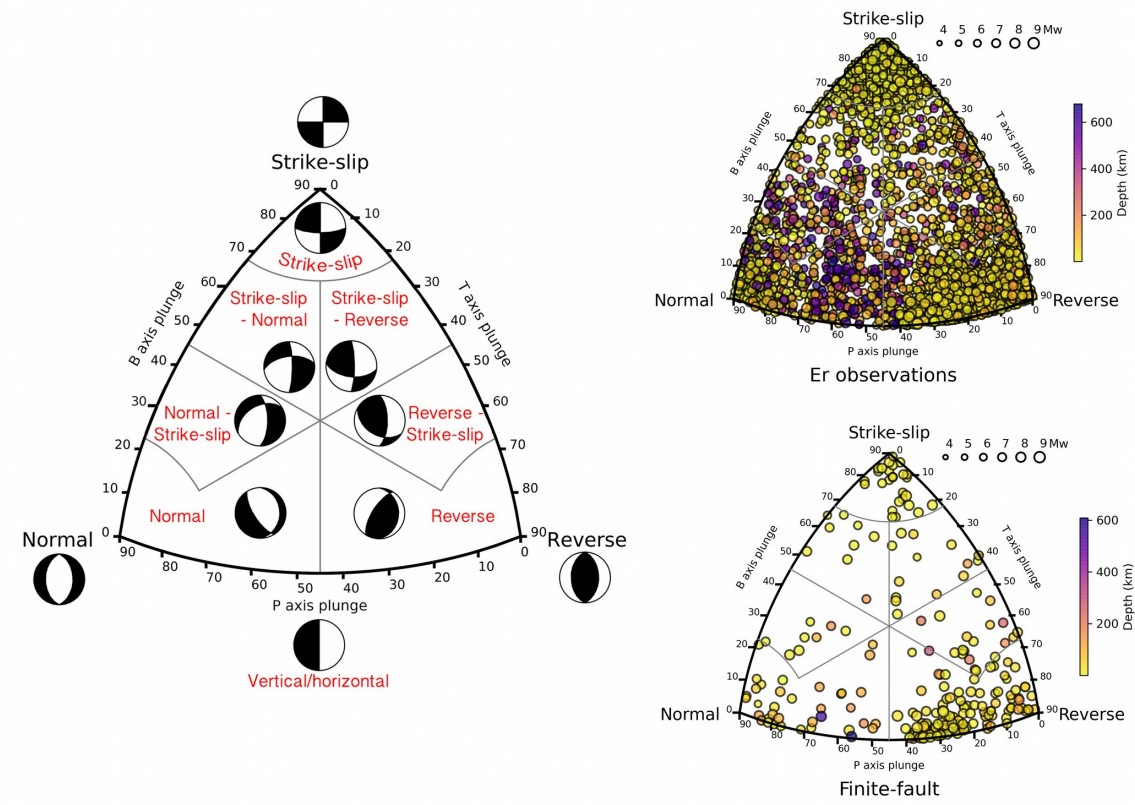

**Figure 1.** The Kaverina fault classification ternary diagram used to classify focal mechanisms (left panel). Focal mechanisms are denoted by circles filled to indicate event depth in km, and the size of the circle indicates the moment magnitude of the earthquake (right panels). The upper right panel shows the rupture type of seismic events with a radiated seismic energy estimation. Rupture type of seismic events with a finite-fault model used to estimate the radiated energy (lower right panel).



## Radiated seismic observations

## Finite−fault models

537

**Figure 2.** Hypocenter location and rupture type classification of earthquakes with reported radiated seismic energy ($E_R$) (upper panel). Hypocenter location and rupture type classification of earthquakes with a finite-fault model used to calculate the radiated seismic energy ($E_R$) (lower panel).R, reverse; R-SS, reverse–strike-slip; SS, strike-slip; SS-R, strike-slip–reverse; SS-N, strike-slip–normal; N, normal; and N-SS, normal–strike-slip.



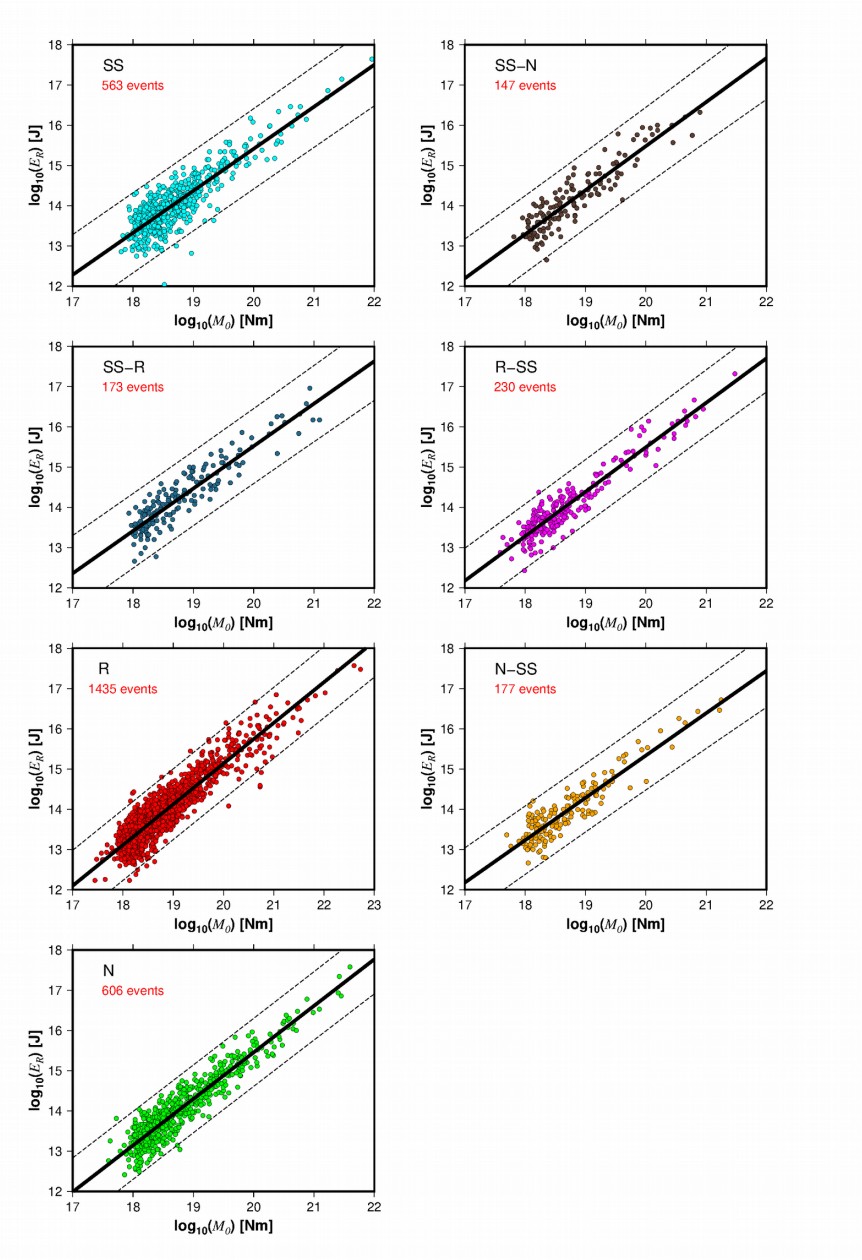

**Figure 3.** The radiated seismic energy ($E_R$) as a function of the seismic moment ($M_0$) for the different rupture types. The solid black lines represent the best fit, and the dashed lines indicate the 95% confidence interval about the regression lines.



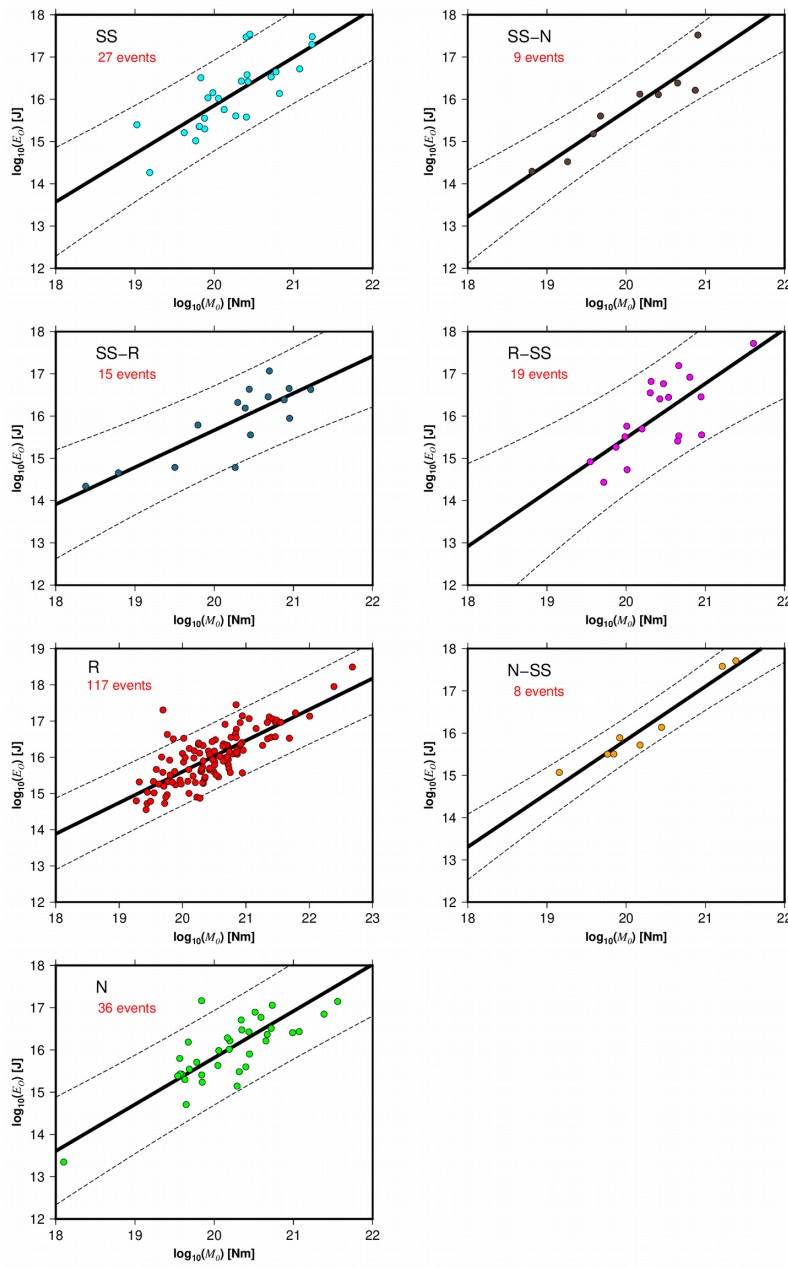

**Figure 4.** The overdamped dynamics approximation of the radiated energy ($E_O$) as a function of the seismic moment ($M_0$) for the different rupture types. The solid black lines represent the best fit, and the dashed lines indicate the 95% confidence interval about the regression lines.



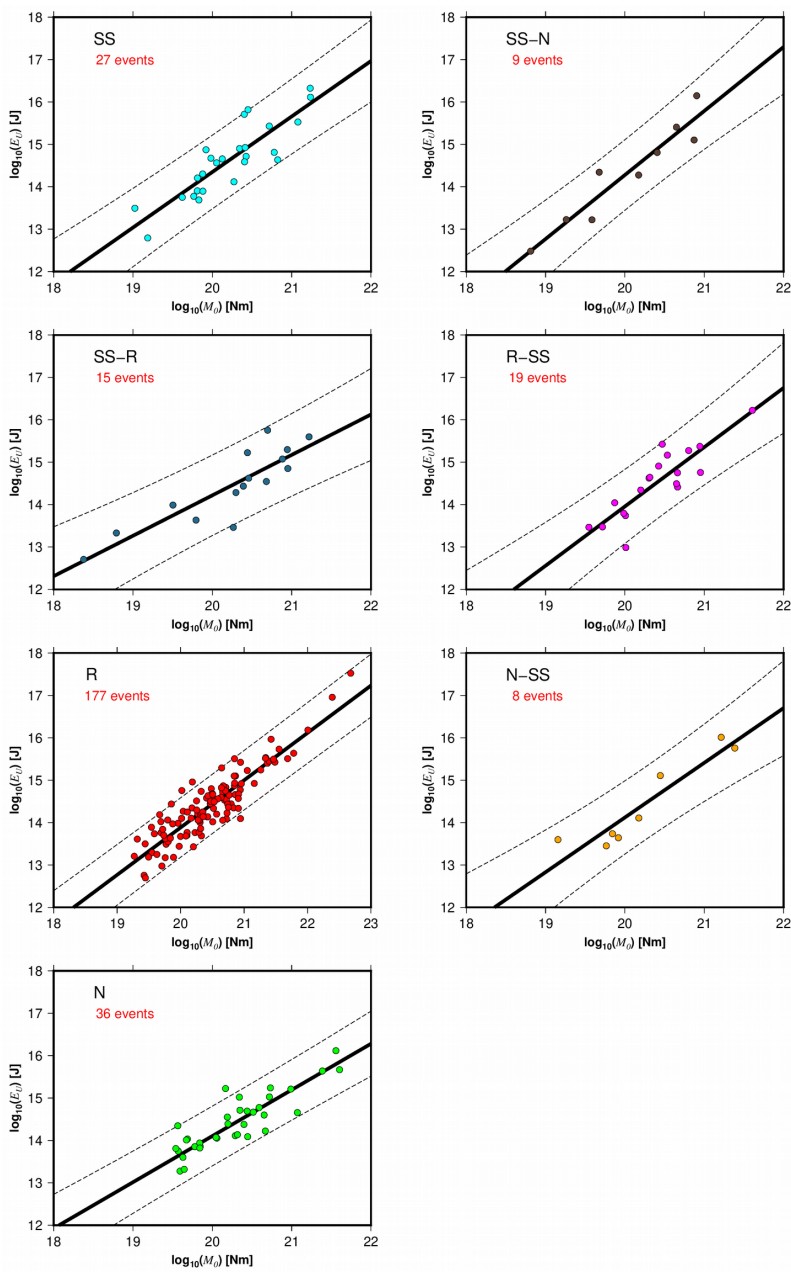

**Figure 5.** The energy obtained from the averaged finite-fault model ($E_U$) as a function of the seismic moment ($M_0$) for the different rupture types. The solid black lines represent the best fit, and the dashed lines indicate the 95% confidence interval about the regression lines.

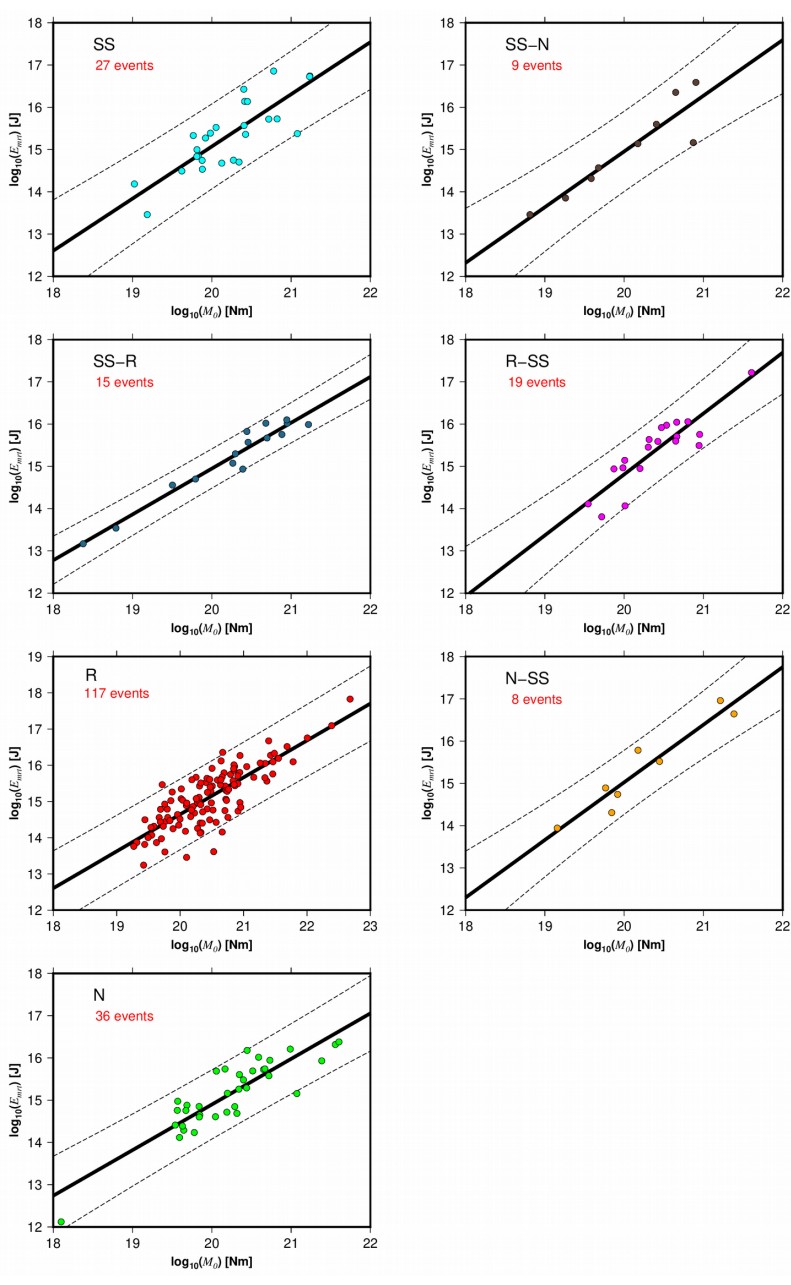

555

**Figure 6.** The radiated seismic energy based on moment rate functions ($E_{rmt}$) versus seismic moment

($M_0$) for the different rupture types. The solid black lines represent the best fit, and the dashed lines

indicate the 95% confidence interval about the regression lines.



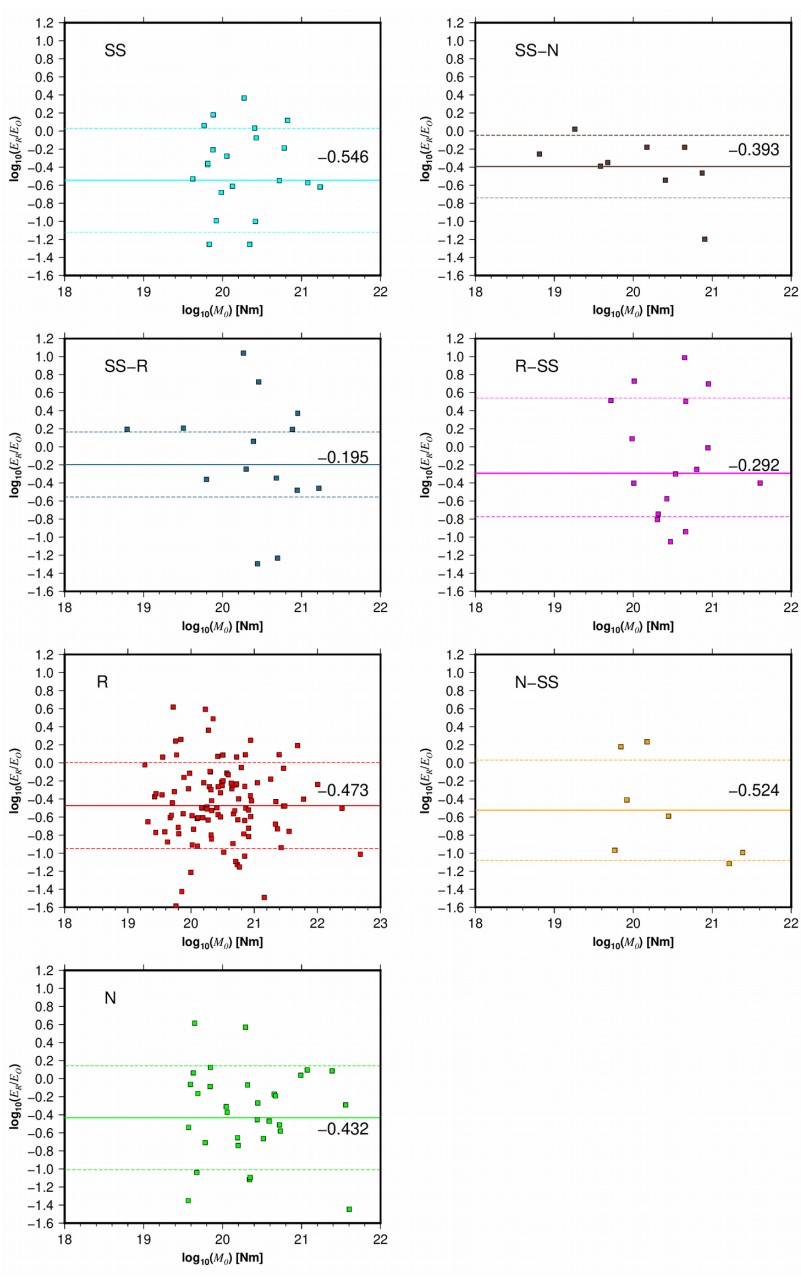

559

**Figure 7.** Comparison between radiated seismic energy based on velocity flux integration ($E_R$) and overdamped ($E_O$) energy estimations. Lines represent the mean values (continuous) of different rupture types and their standard deviation (dashed).



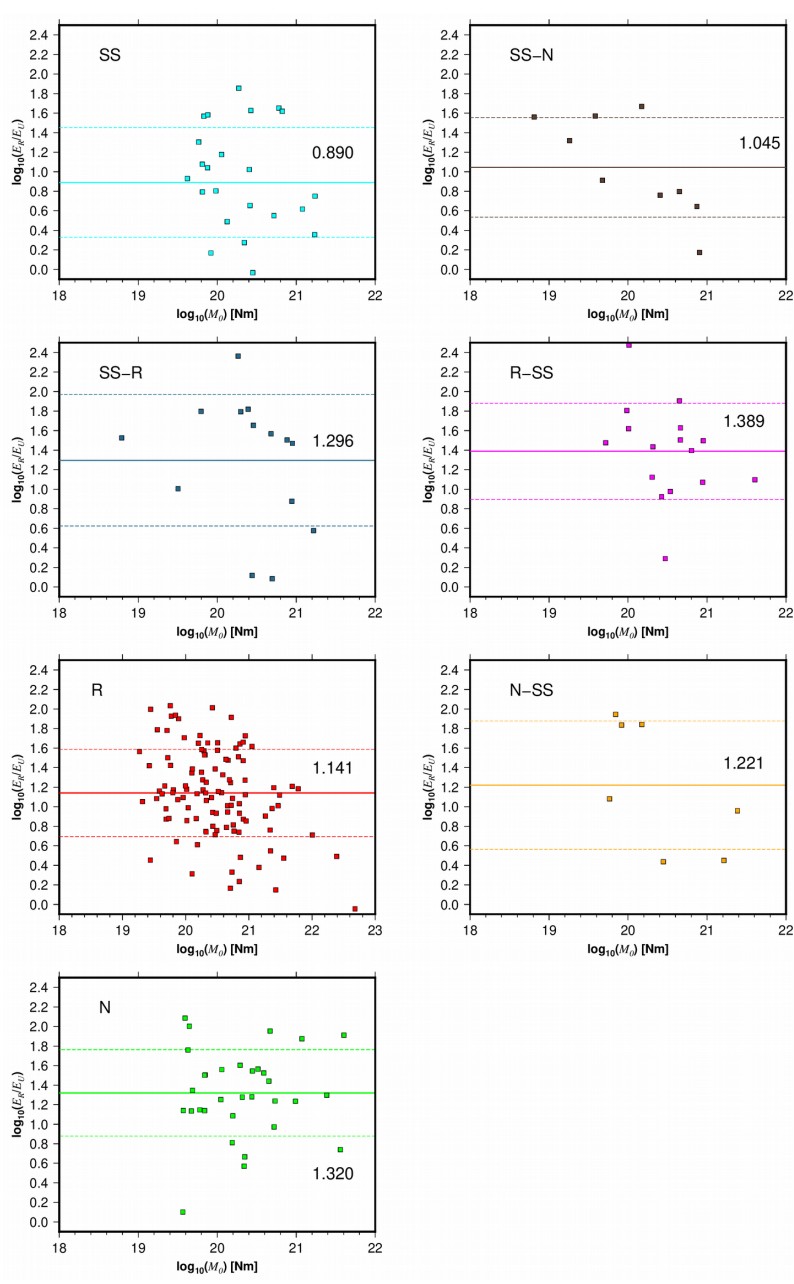

**Figure 8.** Comparison between the ratio of radiated seismic energy based on velocity flux integration ($E_R$) and averaged finite-fault model energy ($E_U$) estimations as a function of seismic moment. Lines represent the mean values (continuous) of different rupture types and their standard deviation (dashed).



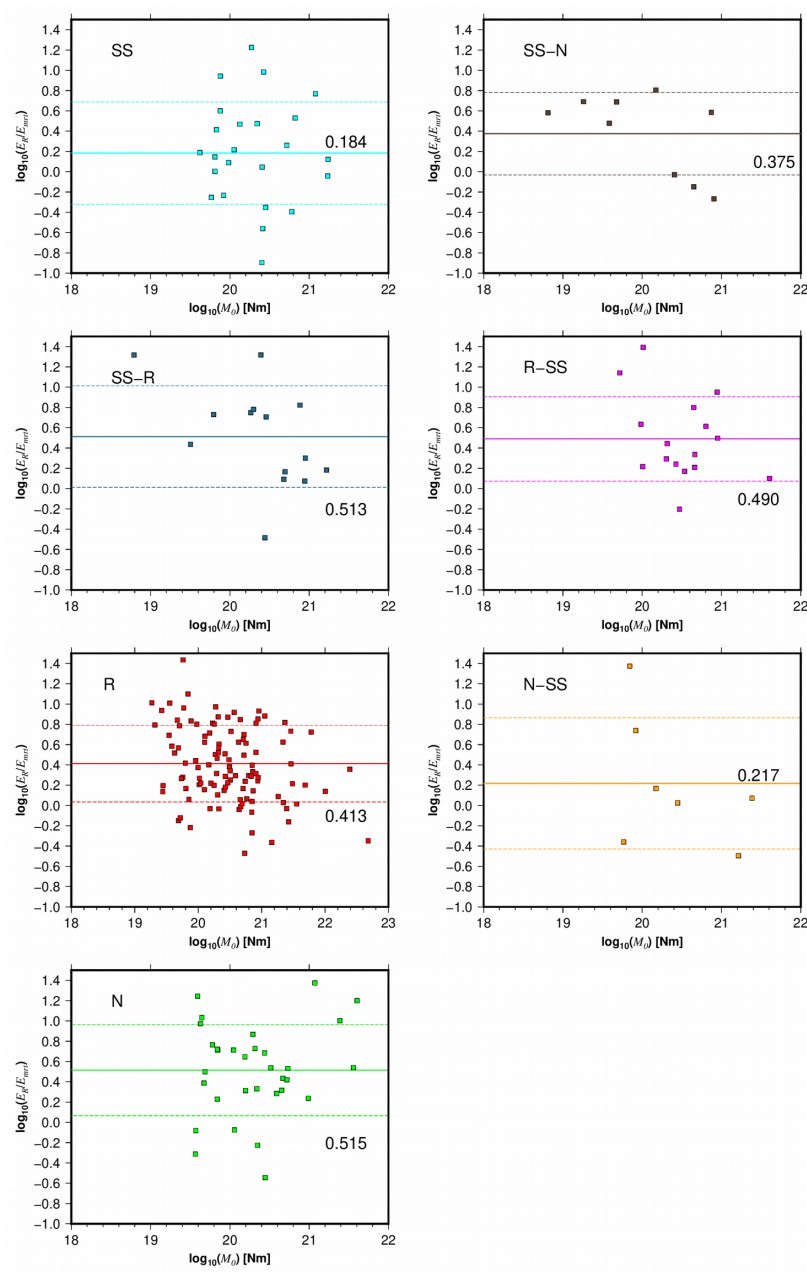

567

**Figure 9.** Comparison between the ratio of radiated seismic energy based on velocity flux integration ($E_R$) and moment rate ($E_{mrt}$) energy estimations as a function of seismic moment. Lines represent the mean values (continuous) of different rupture types and their standard deviation (dashed).





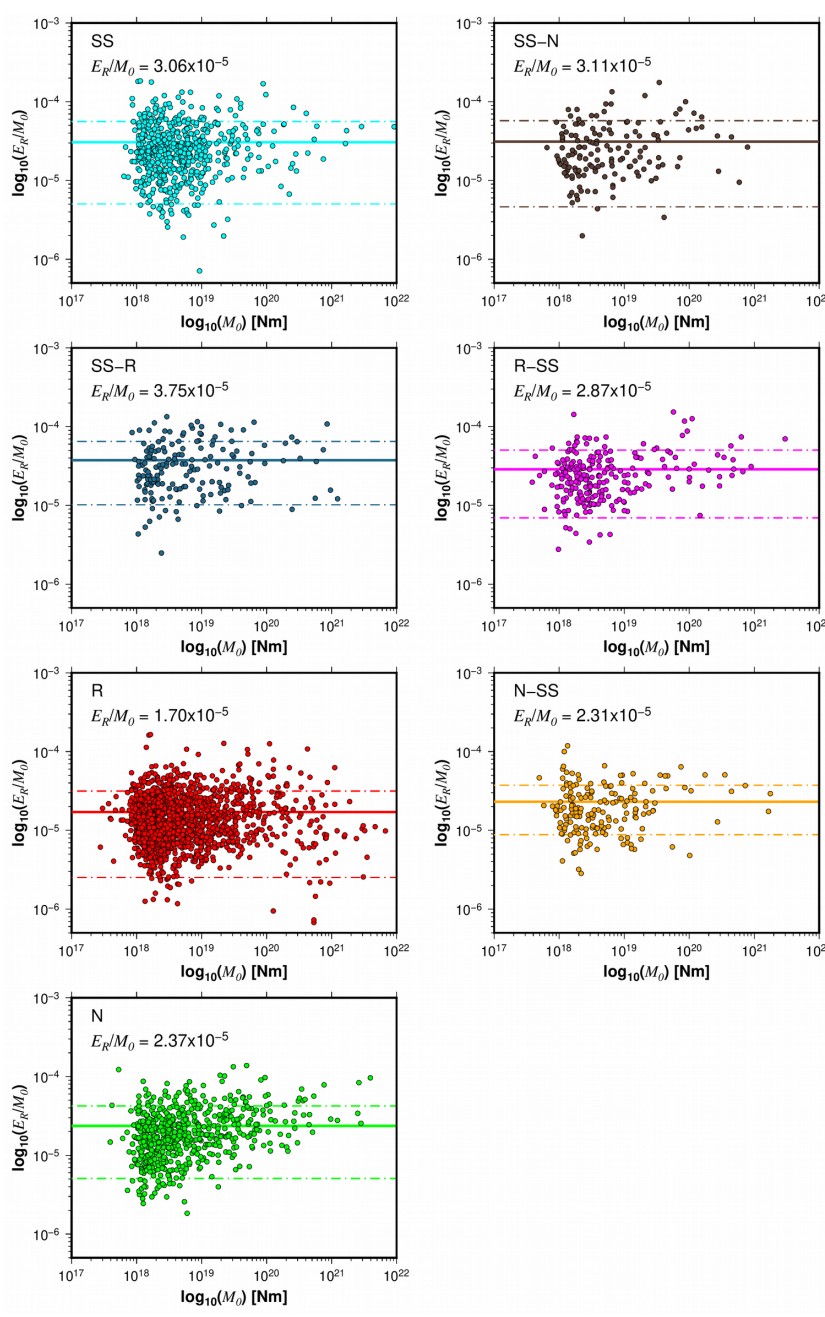

**Figure 10.** The estimated energy-to-moment ratios plotted as a function of the seismic moment for all the rupture types. The solid and dashed lines show the mean value and standard deviations, respectively.





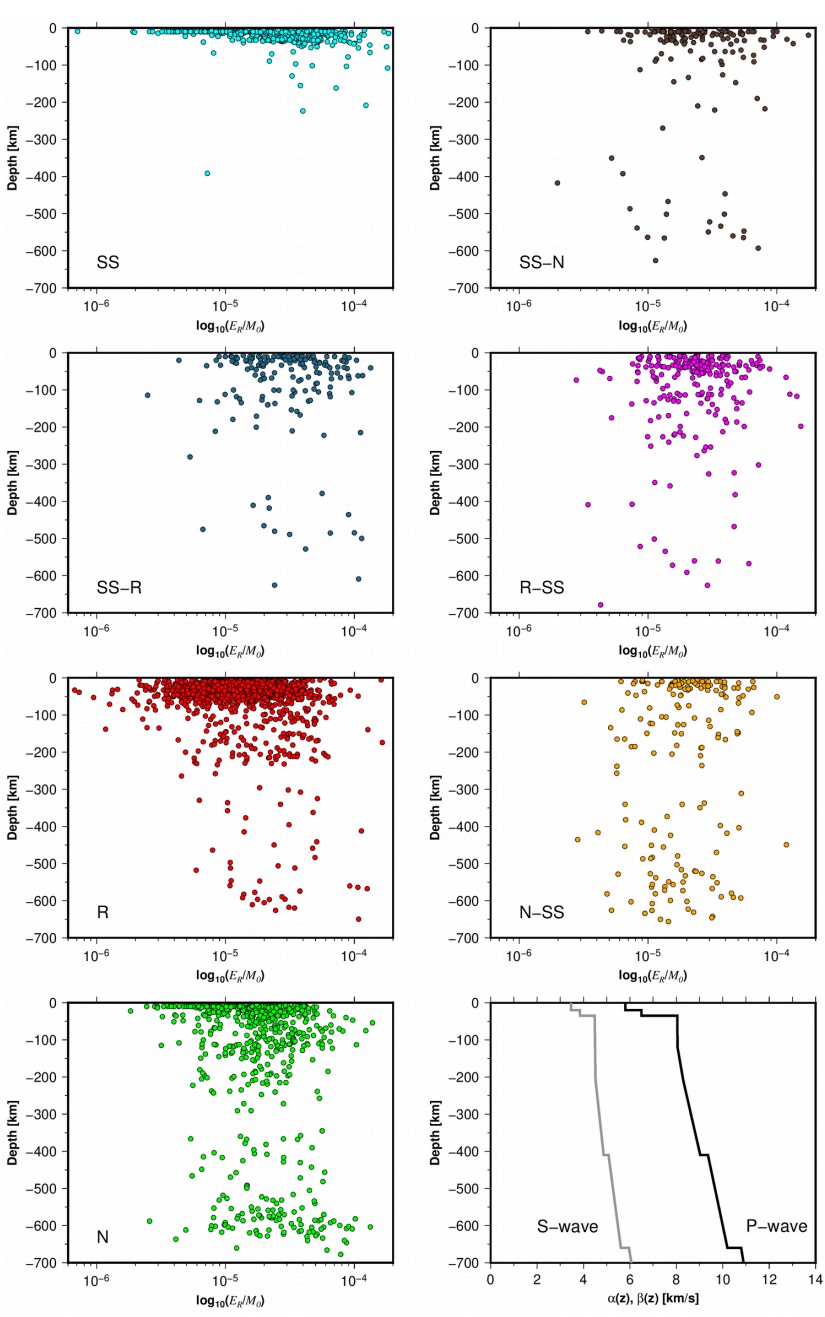

**Figure 11.** The calculated hypocentral depth for all the rupture types as a function of energy-to-moment

ratios. Lower right panel shows the ak135-F global velocity model.



**Figure 12.** Hypocentral depth for the different rupture subsets as a function of apparent stress ($\tau_a$).
Color curves are the probability density functions (PDFs). Calculated rigidity as a function depth based
on the ak135-F global velocity model used to calculate $\tau_a$ (lower right panel).





**Table 1.** Regression results for the radiated seismic energy scaling relationships. The scaling relation is given by $\log_{10} E = a \log_{10} M_0 + b$, where $E$ is the radiated seismic energy based on velocity flux integration ($E_R$), the overdamped dynamics approximation of the radiated energy ($E_O$), the energy obtained from the averaged finite-fault model ($E_U$), or the energy obtained from moment rate functions ($E_{mrt}$) in J, $M_0$ is the seismic moment in Nm. $D^2$ is the determination coefficient, $a$ is the slope, $Sa$ is the standard error of $a$, $b$ is the intercept, and $Sb$ is the standard error of $b$.

| Parameter | $a$ | $Sa$ | $b$ | $Sb$ | $D^2$ | Rupture type | Method |
|---|---|---|---|---|---|---|---|
| $E_R$ [J] | 1.04 | 0.02 | -5.47 | 0.47 | 0.76 | SS | Velocity flux integration |
| $E_R$ [J] | 1.09 | 0.04 | -6.42 | 0.78 | 0.83 | SS-N | Velocity flux integration |
| $E_R$ [J] | 1.05 | 0.03 | -5.57 | 0.65 | 0.84 | SS-R | Velocity flux integration |
| $E_R$ [J] | 1.10 | 0.03 | -6.62 | 0.48 | 0.89 | R-SS | Velocity flux integration |
| $E_R$ [J] | 1.01 | 0.01 | -5.10 | 0.21 | 0.85 | R | Velocity flux integration |
| $E_R$ [J] | 1.05 | 0.03 | -5.72 | 0.64 | 0.84 | N-SS | Velocity flux integration |
| $E_R$ [J] | 1.16 | 0.02 | -7.67 | 0.33 | 0.87 | N | Velocity flux integration |
| | | | | | | | |
| $E_O$ [J] | 1.14 | 0.16 | -6.93 | 3.17 | 0.68 | SS | Finite-fault model |
| $E_O$ [J] | 1.25 | 0.18 | -9.35 | 3.67 | 0.87 | SS-N | Finite-fault model |
| $E_O$ [J] | 0.88 | 0.17 | -1.86 | 3.39 | 0.68 | SS-R | Finite-fault model |
| $E_O$ [J] | 1.28 | 0.30 | -10.21 | 6.18 | 0.51 | R-SS | Finite-fault model |
| $E_O$ [J] | 0.86 | 0.07 | -1.57 | 1.38 | 0.59 | R | Finite-fault model |
| $E_O$ [J] | 1.27 | 0.13 | -9.50 | 2.55 | 0.94 | N-SS | Finite-fault model |
| $E_O$ [J] | 1.10 | 0.14 | -6.26 | 2.80 | 0.65 | N | Finite-fault model |
| | | | | | | | |
| $E_U$ [J] | 1.31 | 0.13 | -11.85 | 2.56 | 0.81 | SS | Finite-fault model |
| $E_U$ [J] | 1.51 | 0.19 | -15.92 | 3.76 | 0.90 | SS-N | Finite-fault model |
| $E_U$ [J] | 0.95 | 0.15 | -4.86 | 3.06 | 0.75 | SS-R | Finite-fault model |
| $E_U$ [J] | 1.40 | 0.20 | -14.00 | 4.05 | 0.74 | R-SS | Finite-fault model |
| $E_U$ [J] | 1.12 | 0.05 | -8.44 | 1.03 | 0.81 | R | Finite-fault model |
| $E_U$ [J] | 1.29 | 0.20 | -11.68 | 4.11 | 0.87 | N-SS | Finite-fault model |
| $E_U$ [J] | 1.09 | 0.09 | -7.68 | 1.76 | 0.82 | N | Finite-fault model |
| | | | | | | | |
| $E_{mrt}$ [J] | 1.23 | 0.15 | -9.61 | 2.97 | 0.74 | SS | Moment rate function |
| $E_{mrt}$ [J] | 1.32 | 0.21 | -11.42 | 4.30 | 0.84 | SS-N | Moment rate function |
| $E_{mrt}$ [J] | 1.08 | 0.07 | -6.75 | 1.50 | 0.94 | SS-R | Moment rate function |
| $E_{mrt}$ [J] | 1.44 | 0.18 | -14.02 | 3.71 | 0.79 | R-SS | Moment rate function |
| $E_{mrt}$ [J] | 1.02 | 0.07 | -5.76 | 1.44 | 0.65 | R | Moment rate function |
| $E_{mrt}$ [J] | 1.36 | 0.18 | -12.25 | 3.61 | 0.91 | N-SS | Moment rate function |
| $E_{mrt}$ [J] | 1.08 | 0.10 | -6.68 | 2.05 | 0.77 | N | Moment rate function |





**Table 2.** Conversion relationships among the different types of energies. $E_R$ is the radiated seismic energy based on velocity flux integration, $E_O$ is the overdamped dynamics approximation of the radiated energy, $E_U$ is the energy obtained from the averaged finite-fault model, and $E_{mrt}$ is the energy obtained from moment rate functions.

| Rupture type | Parameters | Model | $a$ | $Sa$ | $b$ | $Sb$ | $D^2$ |
|---|---|---|---|---|---|---|---|
| SS | $E_R, E_O$ | $\log_{10}E_R = a \log_{10}E_O + b$ | 0.61 | 0.12 | 5.83 | 1.90 | 0.54 |
| SS-N | $E_R, E_O$ | $\log_{10}E_R = a \log_{10}E_O + b$ | 0.75 | 0.09 | 3.60 | 1.42 | 0.91 |
| SS-R | $E_R, E_O$ | $\log_{10}E_R = a \log_{10}E_O + b$ | 0.37 | 0.16 | 9.96 | 2.60 | 0.30 |
| N-SS | $E_R, E_O$ | $\log_{10}E_R = a \log_{10}E_O + b$ | 0.61 | 0.19 | 5.78 | 3.19 | 0.66 |
| N | $E_R, E_O$ | $\log_{10}E_R = a \log_{10}E_O + b$ | 0.59 | 0.10 | 6.23 | 1.67 | 0.52 |
| R-SS | $E_R, E_O$ | $\log_{10}E_R = a \log_{10}E_O + b$ | 0.44 | 0.12 | 8.90 | 1.95 | 0.49 |
| R | $E_R, E_O$ | $\log_{10}E_R = a \log_{10}E_O + b$ | 0.70 | 0.06 | 4.27 | 0.91 | 0.59 |
| | | | | | | | |
| SS | $E_R, E_U$ | $\log_{10}E_R = a \log_{10}E_U + b$ | 0.61 | 0.11 | 6.67 | 1.59 | 0.59 |
| SS-N | $E_R, E_U$ | $\log_{10}E_R = a \log_{10}E_U + b$ | 0.63 | 0.08 | 6.40 | 1.18 | 0.89 |
| SS-R | $E_R, E_U$ | $\log_{10}E_R = a \log_{10}E_U + b$ | 0.35 | 0.17 | 10.73 | 2.43 | 0.28 |
| N-SS | $E_R, E_U$ | $\log_{10}E_R = a \log_{10}E_U + b$ | 0.54 | 0.18 | 7.96 | 2.65 | 0.63 |
| N | $E_R, E_U$ | $\log_{10}E_R = a \log_{10}E_U + b$ | 0.78 | 0.11 | 4.50 | 1.62 | 0.61 |
| R-SS | $E_R, E_U$ | $\log_{10}E_R = a \log_{10}E_U + b$ | 0.56 | 0.11 | 7.82 | 1.58 | 0.66 |
| R | $E_R, E_U$ | $\log_{10}E_R = a \log_{10}E_U + b$ | 0.69 | 0.04 | 5.67 | 0.63 | 0.69 |
| | | | | | | | |
| SS | $E_R, E_{mrt}$ | $\log_{10}E_R = a \log_{10}E_{mrt} + b$ | 0.66 | 0.10 | 5.49 | 1.56 | 0.65 |
| SS-N | $E_R, E_{mrt}$ | $\log_{10}E_R = a \log_{10}E_{mrt} + b$ | 0.70 | 0.09 | 4.93 | 1.32 | 0.90 |
| SS-R | $E_R, E_{mrt}$ | $\log_{10}E_R = a \log_{10}E_{mrt} + b$ | 0.52 | 0.14 | 7.84 | 2.16 | 0.54 |
| N-SS | $E_R, E_{mrt}$ | $\log_{10}E_R = a \log_{10}E_{mrt} + b$ | 0.55 | 0.21 | 7.23 | 3.30 | 0.57 |
| N | $E_R, E_{mrt}$ | $\log_{10}E_R = a \log_{10}E_{mrt} + b$ | 0.78 | 0.11 | 3.81 | 1.79 | 0.60 |
| R-SS | $E_R, E_{mrt}$ | $\log_{10}E_R = a \log_{10}E_{mrt} + b$ | 0.62 | 0.10 | 6.41 | 1.50 | 0.75 |
| R | $E_R, E_{mrt}$ | $\log_{10}E_R = a \log_{10}E_{mrt} + b$ | 0.73 | 0.04 | 4.54 | 0.55 | 0.78 |



**Table 3.** Estimations of average apparent stress ($\tau_\alpha$) for different faulting types based on the velocity
flux integration method. $\tau_\alpha$ is calculated with the following model: $\log_{10}E_R = \log_{10} M_0+b$, where $\tau_\alpha = \mu$
$10^b$. We assume $\mu=\bar{\mu}$ as the average rigidity in a specific depth interval of 30 km. $\tau_\alpha^1$ and $\tau_\alpha^2$ are the
95% de upper and lower confidence intervals for the mean. 3 and 4 indicate $\tau_\alpha$ results from Choy and
Boatwright (1995) and Pérez-Campos and Beroza (2001), respectively (botton lines).

| Depth [km] | $\bar{\mu}$ [MPa] | $\tau_\alpha$[MPa] | | | | | | | $\tau_\alpha^1$[MPa] | | | | | | | $\tau_\alpha^2$[MPa] | | | | | | |
|---|---|---|---|---|---|---|---|---|---|---|---|---|---|---|---|---|---|---|---|---|---|---|
| | | SS | SS-N | SS-R | N-SS | N | R-SS | R | SS | SS-N | SS-R | N-SS | N | R-SS | R | SS | SS-N | SS-R | N-SS | N | R-SS | R |
| 0 ≤ z ≤ 30 | 3.48 x 10⁴ | 0.72 | 0.75 | 0.90 | 0.72 | 0.50 | 0.79 | 0.43 | 3.51 | 3.31 | 3.41 | 2.20 | 1.91 | 2.34 | 1.40 | 0.15 | 0.17 | 0.24 | 0.24 | 0.13 | 0.26 | 0.13 |
| 30 < z ≤ 60 | 5.33 x 10⁴ | 1.95 | 1.49 | 2.47 | 1.33 | 1.03 | 1.29 | 0.68 | 6.76 | 8.65 | 9.79 | 6.55 | 4.57 | 4.92 | 2.82 | 0.56 | 0.26 | 0.62 | 0.27 | 0.23 | 0.39 | 0.16 |
| 60 < z ≤ 90 | 6.65 x 10⁴ | | 1.75 | 3.08 | | 1.58 | 1.37 | 0.73 | | 6.75 | 12.21 | | 6.85 | 9.55 | 4.33 | | 0.45 | 0.78 | | 0.37 | 0.19 | 0.12 |
| 90 < z ≤ 120 | 6.67 x 10⁴ | | 1.88 | | | 1.49 | 1.96 | 1.45 | | 13.59 | | | 5.95 | 8.55 | 7.08 | | 0.26 | | | 0.37 | 0.45 | 0.30 |
| 120 < z ≤ 150 | 6.73 x 10⁴ | | | 1.22 | 1.15 | 1.13 | 1.38 | 0.90 | | | 5.55 | 6.57 | 3.76 | 5.43 | 7.86 | | | 0.27 | 0.20 | 0.34 | 0.35 | 0.10 |
| 150 < z ≤ 180 | 6.81 x 10⁴ | | | | | 1.55 | | 1.38 | | | | | 3.93 | | 7.79 | | | | | 0.61 | | 0.24 |
| 180 < z ≤ 210 | 6.90 x 10⁴ | | | | | 1.09 | | 1.35 | | | | | 4.07 | | 5.52 | | | | | 0.29 | | 0.33 |
| 210 < z ≤ 240 | 7.07 x 10⁴ | | | | | 1.19 | | 1.34 | | | | | 5.17 | | 6.04 | | | | | 0.27 | | 0.30 |
| 540 < z ≤ 570 | 1.16 x 10⁵ | | | | | 2.39 | | | | | | | 7.61 | | | | | | | 0.75 | | |
| 570 < z ≤ 600 | 1.19 x 10⁵ | | | | | 2.88 | | | | | | | 14.88 | | | | | | | 0.56 | | |
| 600 < z ≤ 630 | 1.23 x 10⁵ | | | | | 3.33 | | | | | | | 18.76 | | | | | | | 0.59 | | |
| | 3.00 x 10⁵ | 3.55[3] | | 0.48[3] | | 0.32[3] | | | 20.69[3] | | | | 4.16[3] | | 2.54[3] | 0.61[3] | | | | 0.05[3] | | 0.04[4] |
| | 3.00 x 10⁵ | 0.70[4] | | 0.25[4] | | 0.15[4] | | | 1.01[4] | | | | 0.30[4] | | 0.19[4] | 0.49[4] | | | | 0.21[4] | | 0.12[4] |

**Table 4.** Estimations of average apparent stress ($\tau_\alpha$) for different faulting types based on slip
distributions ($E_{mrt}$, $E_U$, and $E_O$). $\tau_\alpha$ is calculated with the following model: $\log_{10}E_R = \log_{10} M_0+b$, where $\tau_\alpha$
$= \mu 10^b$. We assume $\mu=\bar{\mu}$ as the average rigidity in a specific depth interval of 30 km. $\tau_\alpha^1$ and $\tau_\alpha^2$ are
the 95% de upper and lower confidence intervals for the mean. 3 and 4 indicate $\tau_\alpha$ results from Choy
and Boatwright (1995) and Pérez-Campos and Beroza (2001), respectively (botton lines).

| Depth [km] | $\bar{\mu}$ [MPa] | $\tau_\alpha$[MPa] | | | | | | | $\tau_\alpha^1$[MPa] | | | | | | | $\tau_\alpha^2$[MPa] | | | | | | |
|---|---|---|---|---|---|---|---|---|---|---|---|---|---|---|---|---|---|---|---|---|---|---|
| | | SS | SS-N | SS-R | N-SS | N | R-SS | R | SS | SS-N | SS-R | N-SS | N | R-SS | R | SS | SS-N | SS-R | N-SS | N | R-SS | R |
| **$E_{mrt}$** | | | | | | | | | | | | | | | | | | | | | | |
| 0 ≤ z ≤ 30 | 3.48 x 10⁴ | 0.52 | | 0.33 | | 0.31 | | 0.16 | 5.72 | | 1.36 | | 2.10 | | 1.47 | 0.05 | | 0.08 | | 0.05 | | 0.02 |
| 30 < z ≤ 60 | 5.33 x 10⁴ | | | | | | 0.24 | | | | | | | 2.28 | | | | | | | 0.03 | |
| **$E_U$** | | | | | | | | | | | | | | | | | | | | | | |
| 0 ≤ z ≤ 30 | 3.48 x 10⁴ | 2.78 | | 1.41 | | 2.59 | | 1.50 | 32.77 | | 23.19 | | 21.79 | | 19.92 | 0.24 | | 0.08 | | 0.10 | | 0.11 |
| 30 < z ≤ 60 | 5.33 x 10⁴ | | | | | | 2.31 | | | | | | | 30.51 | | | | | | | 0.17 | |
| **$E_O$** | | | | | | | | | | | | | | | | | | | | | | |
| 0 ≤ z ≤ 30 | 3.48 x 10⁴ | 0.10 | | 0.04 | | 0.04 | | 0.03 | 0.91 | | 0.51 | | 0.24 | | 0.17 | 0.01 | | 0.01 | | 0.09 | | 0.005 |
| 30 < z ≤ 60 | 5.33 x 10⁴ | | | | | | 0.04 | | | | | | | 0.25 | | | | | | | 0.007 | |
| | 3.00 x 10⁵ | 3.55[3] | | 0.48[3] | | 0.32[3] | | | 20.69[3] | | | | 4.16[3] | | 2.54[3] | 0.61[3] | | | | 0.05[3] | | 0.04[4] |
| | 3.00 x 10⁵ | 0.70[4] | | 0.25[4] | | 0.15[4] | | | 1.01[4] | | | | 0.30[4] | | 0.19[4] | 0.49[4] | | | | 0.21[4] | | 0.12[4] |
