# Peer review of "Global seismic energy scaling relationships based on the type of faulting."

_EGUsphere, 2023_

## Author Comment (AC1)

**Response to comments by the reviewers and editor**

We appreciate the comments from the reviewers, which have allowed us to improve the manuscript. Overall, we have followed all the suggestions. We now give a response to the individual points raised. Changes made to the manuscript are highlighted in yellow color.

**Reviewer 1.**

**I appreciated the quality of this manuscript, which presents the results of analysis of the ratio between radiated energy and seismic moment of earthquakes taking into account different focal mechanism, depths and seismic regions. The results are expressed in terms of stress drop on the source. The authors state that the results and their conclusions can be affected by uncertainties on the observation data and the used model. In this respect, I would suggest to add a statistical analysis of the results in order to quantify the probability of rejecting the null hypothesis that the differences among the various cases may be caused just by random variations of the observations.**

We statistically analyze the different energy measurements considering different rupture types and depths. The results and tables of the statistical analysis are presented in the version of the manuscript.

**Reviewer 2.**

**This manuscript describes the detailed work by the authors to investigate energy released by earthquakes globally. They compare energy obtained through various approaches and investigate systematic variation of energy to moment ratio for different fault types and hypocentral depths. Furthermore, they develop conversion relationships between the different energy estimates. The work is consistent with previous studies, but a valuable contribution as a consistent approach is used to all earthquakes that are part of the study. The paper is well written (although consistency of tense should be checked) and the application of methods and results are sound. However, one issue is that it remains unclear whether the authors computed energy themselves - my understanding is that ER is taken from IRIS (which is fine), but that the other energy values are computed in this study - this should be clearer throughout the text.**

**I have the following more detailed comments that may help to improve the manuscript:**

**line 1: title - 'based on' does not seem correct, change to e.g. '...: Investigating dependence on type of faulting and hypocentral depth'**

We decided to keep the title as present in this manuscript.

**line 39: please check, what is difference between first and second radiated energy in this sentence?**

We clarified this point in the manuscript.

**line 53: dependence of ER**

We corrected this sentence.

**line 54: insert 'but' after ','**

We corrected this sentence.

**line 110: gCMT needs to be explained**

We explained gCMT in the manuscript.

**line 116: Do I understand correctly that ER is taken from IRIS, and that EU and EO are computed in this study? This should be expressed more clearly throughout the text.**

That is correct.

**line 119: in case the results are taken from IRIS, this section is too detailed.**

We decided to keep this section as it is because it explains the method used to calculate Er.

**line 120: change to 'we describe' - please check tense throughout the text, I give some more examples where I think it can be changed, but not for everything**

We corrected this sentence.

**line 121: which study? not clear so far if new calculations are done in this paper, or if results are taken from IRIS; if taken from IRIS perhaps not all equations are needed, unless important for the discussion.**

We clarified this point in the text and we considered that equations are important for the discussion.

**line 185: show**

We corrected this sentence.

**line 195: show**

We corrected this sentence.

**line 206: insert apparent stress before tau (even though it is given above)**

We rewrote this sentence.

**line 223: remove 'for strike-slip earthquakes' (as SS is given at start of sentence)**

We corrected this sentence.

**line 224: use 'higher depth'?**

We rewrote the sentence.

**line 256: spell out symbol in ER E (Eu, Eo)**

We corrected the sentence.

**line 267: tsunami earthquake (rather than event)?**

We corrected the sentence.

**line 271: is 'seismogenic' correct here?**

Yes.

**line 276: asperities or asperity patches**

We corrected the sentence.

**line 281: show**

We corrected the sentence.

**line 341: you focus on highest values as they are important in hazard context, but lowest values could also be interesting? overall, you could mention importance of results for seismic hazard**

We followed this suggestion.

**line 362: 'different types of faulting'**

We corrected the sentence.

**line 377: Our higher ER/M0 .…**

We clarified this point in the sentence.